# Exploring Factors Influencing Patient Delay Behavior in Oral Cancer: The Development of a Risk Prediction Model in Western China

**DOI:** 10.3390/healthcare12222252

**Published:** 2024-11-12

**Authors:** Yuanyuan Yang, Huan Ning, Bohui Liang, Huaming Mai, Jie Zhou, Jing Yang, Jiegang Huang

**Affiliations:** 1Hospital of Stomatology, Guangxi Medical University, Nanning 530021, China; yuanyang0122@outlook.com (Y.Y.); huamingmai@outlook.com (H.M.); 2School of Public Health, Guangxi Medical University, Nanning 530021, China; a15277099623@hotmail.com (H.N.); zhoujie2121@outlook.com (J.Z.); jingyang202220919@outlook.com (J.Y.); 3School of Computer, Electronics and Information, Guangxi University, Nanning 530004, China; hliangcs@foxmail.com; 4Guangxi Colleges and Universities Key Laboratory of Prevention and Control of Highly Prevalent Diseases, Guangxi Medical University, Nanning 530021, China

**Keywords:** oral cancer patients, patient delay, influencing factors, prediction model, Guangxi Province, China’s western region, spatial accessibility, machine learning

## Abstract

Background and Aims: To study the unknown influencing factors of delayed medical treatment behavior in oral cancer patients in western China and to develop a prediction model on the risk of delayed medical treatment in oral cancer patients. Method: We investigated oral cancer patients attending a tertiary Grade A dental hospital in western China from June 2022 to July 2023. The logistic regression and four machine learning models (nearest neighbors, the RBF SVM, random forest, and QDA) were used to identify risk factors and establish a risk prediction model. We used the established model to predict the data before and after the COVID-19 pandemic and test whether the prediction effect can still remain stable and accurate under the interference of COVID-19. Result: Out of the 495 patients included in the study, 122 patients (58.65%) delayed seeking medical treatment before the lifting of the restrictions of the pandemic, while 153 patients (53.13%) did so after the lifting of restrictions. The logistic regression model revealed that living with adult children was a protective factor for patients in delaying seeking medical attention, regardless of the implementation of pandemic control measures. After comparing each model, it was found that the statistical indicators of the random forest algorithm such as the AUC score (0.8380) and specificity (0.8077) ranked first, with the best prediction performance and stable performance. Conclusions: This study systematically elucidates the critical factors influencing patient delay behavior in oral cancer diagnosis and treatment, employing a comprehensive risk prediction model that accurately identifies individuals at an elevated risk of delay. It represents a pioneering large-scale investigation conducted in western China, focusing explicitly on the multifaceted factors affecting the delayed medical treatment behavior of oral cancer patients. The findings underscore the imperative of implementing early intervention strategies tailored to mitigate these delays. Furthermore, this study emphasizes the pivotal role of robust social support systems and positive family dynamics in facilitating timely access to healthcare services for oral cancer patients, thereby potentially improving outcomes and survival rates.

## 1. Introduction

Oral cancer is the general term for malignant tumors occurring in the mouth, which are one of the common malignant tumors in the head and neck. Head and neck cancer (HNC) is considered the sixth most common malignant tumor in the world. The incidence and prevalence of oral cancer have significant geographical differences, and their distribution varies greatly around the world. Reports show that Asia has the highest proportion of confirmed cases of oral cancer (64.2%) and mortality (73.3%) [1,2]. Patient delay is defined as the period between the initial recognition of symptoms by the patient and their first consultation with healthcare professionals regarding those symptoms. For tumor diseases, disease staging serves as a crucial prognostic indicator for predicting patient survival. Allison and colleagues discovered that delays exceeding 3 months could elevate the risk of the disease progressing to an advanced stage [3]. Owing to the lack of public awareness regarding oral cancer and inadequate attention to disease symptoms, patients with oral cancer generally experience delays in seeking treatment, which can impact subsequent treatment and recovery outcomes. Currently, research on the delay in seeking medical attention among oral cancer patients in China is scarce, and numerous potential risk factors affecting the medical behavior of these patients remain undiscovered. Public service accessibility is an important research object in urban and rural planning and urban geography abroad. Reachability methods have proposed standards and methods for identifying areas with shortages of public services. It is an important indicator for measuring the level of equalization of urban and rural public services and provides scientific references for the planning and layout of public services [4]. The low level of accessibility to medical and health services leads to patients having to delay seeking medical treatment. Consequently, this study employs the shortest time distance, grounded in GIS (geographic information systems), to delineate the level of accessibility to medical and health services. This approach facilitates a comparison of the disparities in accessibility to medical and health services between the residences of patients who experienced delays in seeking medical treatment and those who did not.

The machine learning predictive model is a tool that utilizes computer algorithms to learn model parameters from training data, which are then used to predict and analyze unknown data. These models are capable of predicting a continuous outcome or classification label based on the input data features. The construction of predictive models typically involves selecting appropriate machine learning algorithms adjusting the model parameters through a training dataset to achieve an optimal predictive performance. The performance of the model is usually evaluated using metrics such as accuracy, precision, and recall to ensure that the model has good generalization ability and stability. For example, Zhang Yuying and others used machine learning algorithms to build a medical delay risk prediction model for imported malaria cases in Jiangsu Province, thereby discovering new risk prediction factors with good predictive efficacy, which can provide a reference for the health management of imported malaria patients. The factors causing delayed medical treatment in oral cancer patients are complex. If machine learning methods can be used to build a risk prediction model for delayed medical treatment in oral cancer, it will fully tap into data information, handle complex interactions between variables, and process various data formats in dynamic, large-capacity, and complex data environments. Thus, it may identify data trends and patterns that could be overlooked in the risk factors of delayed medical treatment for oral cancer, providing guidance for the prevention of the occurrence and development of oral cancer diseases. However, there have been no reports on machine learning prediction models for delayed medical treatment in oral cancer patients in China.

Public service accessibility is an important research object in foreign urban and rural planning and urban geography. The accessibility method proposes standards and methods for identifying public service shortage areas. It is an important indicator for measuring the level of equalization of public services in urban and rural areas and provides a scientific reference for the planning and layout of public services. The low level of accessibility to medical and health services leads to patients having to delay seeking medical treatment. Therefore, this study is based on a geographic information system (GIS) and uses the shortest time distance to represent the level of accessibility to medical and health services, thereby comparing the differences in the accessibility levels of medical and health services between the residences of patients with delayed medical treatment and those without delayed medical treatment.

The western region of China is an economically underdeveloped region with multiple ethnic groups living together. It has its uniqueness in geography, economy, culture, and medical and health resources. In a recent review [5], researchers included studies on the diagnosis delay of oral cancer conducted in multiple countries from 1994 to 2020. Among them, two studies in China were conducted in Beijing and Jilin (a province in northeast China), with sample sizes of 102 and 128, respectively. Large-sample studies on the western region of China have not yet been reported.

The significance of this study lies in that this study is the first large-sample research on the risk factors of delayed medical-seeking behavior among oral cancer patients in western China. In order to explore new factors related to the delayed medical treatment behavior of oral cancer patients, we investigated various aspects, including the patients’ demographic information, their social support status, the site of onset, the impact of the COVID-19, personal health literacy, and the accessibility level of medical and health services in the patients’ place of residence. We also established a new predictive model that can avoid the interference of the pandemic and accurately predict the risk factors for delayed medical treatment in oral cancer patients. This model can be applied to guide the formulation of public health policies, improve the medical treatment behavior of oral cancer patients, and enhance patient survival rates.

## 2. Materials and Methods

### 2.1. Subject Criteria

A continuous inclusion method was used to investigate eligible oral cancer patients who visited the Department of Oral and Maxillofacial Surgery in the Affiliated Stomatological Hospital of Guangxi Medical University from June 2022 to July 2023. This study was conducted at the Affiliated Stomatological Hospital of Guangxi Medical University, which is the only Grade III Class A stomatological specialty hospital in Guangxi, and its oral cancer case data have a certain representativeness. The inclusion criteria: ① suffering from malignant tumors in the oral region, including cancers in the tongue, lips, cheeks, gums, floor of the mouth (oral floor), hard palate, soft palate, and salivary glands; ② pathologically diagnosed with squamous cell carcinoma; ③ age ≥ 18 years old; ④ willing to participate in this study; ⑤ having a certain level of understanding and language expression ability and able to cooperate with the investigation. The exclusion criteria: ① non-permanent residents of Guangxi; ② incomplete medical records. A total of 495 patients met the criteria.

### 2.2. Research Methods

In this study, the delay in seeking medical treatment for oral cancer was defined as the time from the onset of symptoms to the first visit > 3 months. The main form of investigation was face-to-face interviews or telephone follow-ups to conduct questionnaires. The research variables included the patient’s gender, ethnicity, occupation, age, education level, average monthly income, marital status, and inhabiting status, as well as whether they had medical staff as relatives or friends, their type of medical insurance, their systemic disease status, the oral site of symptoms, the initial visit medical institution, self-medication, the impact of the COVID-19 pandemic on visits, smoking and drinking habits, whether they ate betel nuts, personal health literacy (HL), the cognition of the disease, oral health behavior, etc. Furthermore, based on the distribution of medical and health service institutions surrounding the patient’s residence and the type of the nearest road, the level of accessibility to medical and health services was ascertained. The above 21 data types were used as independent variables. Electronic medical records were analyzed, and patient histories were collected. According to the patient’s delay in seeking medical treatment (≤3 months and >3 months), it was used as the dependent variable. Questionnaire information was double-entered using the software EpiData 3.1 and difference tests were performed. Difference samples were rechecked against the original survey data and re-entered.

This paper uses GIS network analysis to calculate the shortest time distance from the patient’s residence to the surrounding primary hospitals, secondary hospitals, tertiary hospitals, dental clinics, and stomatological hospitals, using the shortest time distance to characterize the level of medical and health service accessibility. POI data sources are Gaode Map data from December 2022, including county towns, townships, streets, villages, and urban and rural public service facilities. The road network data is sourced from the OpenStreetMap website, and the traffic network data includes national highways, provincial highways, county roads, township roads, urban arterial roads, secondary arterial roads, and residential area roads. Under the mode of car travel, the speed limits for these roads are 65, 50, 40, 30, 65, 40, and 30 km/h, respectively. The data source for the medical institutions is the Guangxi Medical Insurance Public Service website. The accessibility of medical and health services is divided into four levels, 10~15, 10~20, 20~30, and more than 30 min, which are set as very good, good, poor, and very poor, respectively.

### 2.3. Statistical Analysis Methods

#### 2.3.1. Impact Factor Analysis

Statistical analysis is performed using SPSS 23.0 software. The chi-square test is used for inter-group count data difference analysis, and continuity corrections or Fisher’s exact probability method is used if necessary. After passing the single sample Kolmogorov–Smirnov test, two independent samples’ *t*-tests are used for comparison between groups for measurement data that conforms to a normal distribution. Non-parametric tests are used for comparisons of inter-group differences in non-normally distributed measurement data. *p* < 0.05 indicates a statistically significant difference.

#### 2.3.2. Prediction Model Establishment

Before and after the lifting of the restrictions of the pandemic, patients were randomly assigned to training and test sets in an 8:2 ratio. Python 3.8 software was applied to screen candidate factors in the training sets of the two groups before and after the lifting of the restrictions of the pandemic through the recursive feature elimination (RFE) algorithm. The importance of each feature was obtained by returning the coefficient attribute or feature importance attribute of the learner. RFE was proposed by Guyon et al. [6], based on support vector machines. This method uses a base model for multiple rounds of training. After each round of training, several features with certain weight coefficients are eliminated, and the next round of training is performed based on the new feature set. The implementation steps of RFE in this paper can be summarized as follows: Train a machine learning base model, which is logistic regression in this study. Calculate the importance of different factors in patient delay, eliminate factors with low importance, and repeatedly recurse this step on the feature set until the desired number of features is reached. The most important factors are selected separately in the two groups. A Venn diagram is used to analyze unique or shared factors before and after the lifting of the restrictions of the pandemic.

The above common factors are incorporated into the model to build the overall model. All research subjects are randomly assigned to training sets and test sets in an 8:2 ratio. Risk prediction models are constructed based on four machine learning algorithms: nearest neighbors, the RBF SVM, random forest, and QDA. Ten-fold cross-validation (Figure 1) is used to determine the optimal hyperparameters of the model, which is independently validated on the test set to obtain the model with the highest AUC. The built model is used to predict data before and after the pandemic to test whether the prediction effect is stable and accurate. This process is shown in Figure 2.

The Hospital of Stomatology, Guangxi Medical University, Institutional Review Board provided ethics approval (ethics approval number: 2023006). Written informed consent was obtained from all study participants at the time of initial enrolment. All methods were performed in accordance with the relevant guidelines and regulations.

## 3. Results

### 3.1. Baseline Data Analysis

In this survey, a total of 495 cases of oral cancer in Guangxi were included. The tumor type the participants suffered from, the TNM staging, and whether there were systemic diseases are shown in Table 1. Before the lifting of the pandemic restrictions, there were 208 patients, of whom 122 (58.65%) had delayed seeking medical attention. After the lifting of the pandemic restrictions, there were 287 patients, of whom 153 (53.13%) had delayed seeking medical attention. The demographic characteristics are shown in Table 2. It was found that there were differences in occupation and marital status before and after the relaxation of the pandemic prevention and control. The proportion of workers and farmers decreased after the relaxation, while the proportion of single people increased (Table 2).

The univariate analysis found that before the lifting of the pandemic restrictions, the factors with statistical differences between patients with delayed medical attention and non-delay patients were as follows: sites inside the mouth, heath literacy level, and ways to access knowledge about oral cancer. After the lifting of the restrictions of the pandemic, statistically significant factors associated with patient delay in seeking medical treatment compared to non-delayed patients were having or not having medical as staff friends/relatives and the habit of chewing hard objects (Appendix A).

Regardless of whether it was before or after the lifting of the restrictions of the pandemic, statistically significant differences were observed between patients who delayed seeking medical treatment and those who did not in terms of inhabiting status, urban or rural residence, the impact of the COVID-19 pandemic, the personal health literacy scale, regular oral examination habits, the frequency of tooth brushing, the distance to the nearest medical institution, and the mode of transportation used for medical visits (*p* < 0.05).

The binary logistics regression model indicated that living with adult children was a protective factor in both the periods before and after the lifting of the restrictions of the pandemic. Walking to seek medical care was a risk factor. Before the lifting of the restrictions of the pandemic, being affected by COVID-19, not having regular dental check-up habits, and acquiring knowledge from the internet or other sources were risk factors for patient delay. After the lifting of the restrictions of the pandemic, the habit of biting hard objects with the teeth was a risk factor for affected patients’ medical behavior. Patients living within 2–10 km of the nearest medical institution were at a higher risk of patient delay (Table 3).

### 3.2. Spatial Accessibility Analysis of Medical and Health Services for Oral Cancer Patients in Guangxi

Oral cancer patients seeking medical care are distributed across various cities in Guangxi, radiating from Nanning City as the center (Figure 3A). The proportion is higher in the southeastern part and lower in the northwestern part. This is related to the location of our hospital in Nanning, which is in the central–southern part of Guangxi, and the relatively distant distance from Nanning to the northwestern region of Guangxi, indicating that spatial distance has a certain impact on patients’ medical behavior.

Based on GIS, the shortest time distance was used to represent the accessibility level of medical and health services. Under the mode of automobile travel, the accessibility level of primary hospitals is the highest (Figure 3B): the proportion of patients arriving at first-level institutions within 10 min and 20 min reaches 79.5% and 92.3%, respectively. This indicates that under the background of urban–rural transportation integration, the network of primary medical and health institutions in Guangxi has covered rural areas, making it more convenient for urban and rural residents in Guangxi to seek medical care.

The accessibility level of second-level medical institutions is relatively good (Figure 3C), with the proportion of patients arriving at secondary hospitals within 10 min and 20 min being 55.7% and 71.3%, respectively. The accessibility level of tertiary hospitals is relatively poor (Figure 3D), with only 36.5% and 45.3% of patients able to reach these institutions within 10 min and 20 min, respectively. The regions with better accessibility are mostly urban areas, which may be related to the fact that tertiary medical institutions are mostly built in densely populated urban areas.

The accessibility level of stomatological hospitals is the worst (Figure 3E), with only 17.4% of patients able to arrive within 20 min. There are only two stomatological hospitals in Guangxi, one in Nanning and the other in Guilin, reflecting that residents in Nanning and Guilin enjoy better medical resources. Fortunately, even though the distribution of stomatological hospitals is scarce, oral cancer patients can first go to the stomatology department of general hospitals or dental clinics for screening. The accessibility level of dental clinics is average (Figure 3F). The proportions of patients who can reach a dental clinic within 10 min and 20 min are 46.0% and 55.6%, respectively. However, dental clinics are mostly located in densely populated urban areas and rarely in remote rural areas.

### 3.3. Overall Model Establishment

Feature selection was performed using machine learning methods to remove irrelevant and redundant features and retain representative informative features. In this study, a feature selection method based on the recursive feature elimination (RFE) of logistic regression was used to select factors in the data before and after the lifting of the restrictions of the pandemic (Figure 4). Ten factors were selected before the relaxation: marital status, living with others, the person lived with, the follow-up status of systemic disease, the impact of COVID-19, HL2, HL3, access to knowledge, checking the mouth regularly, and the nearest medical facility. Eighteen factors were selected after the relaxation: living with others, the person lived with, medical staff as friends/relatives, medical insurance type, the type of self-medication, the impact of COVID-19, smoking dosage, second-hand smoking, the consumption of areca nuts, HL1, HL2, HL3, HL score, heath literacy level, presumed cause, checking the mouth regularly, brushing frequency, and the accessibility of secondary care facilities.

Venn diagrams were used to analyze unique or shared factors before and after the lifting of the restrictions of the pandemic. As shown in Figure 5, six shared factors were identified: checking the mouth regularly, HL2, HL3, the impact of COVID-19, living with others, and the person who lives with them. These six factors were included in all the study subjects, and four models were constructed. The hyperparameters of the four machine learning models were adjusted through ten-fold cross-validation, and the optimal hyperparameter combination was determined from each model with the best predictive performance (Figure 6). The ROC curve AUC values of the models were ranked from high to low using random forest (0.838), the RBF SVM (0.809), nearest neighbors (0.787) and QDA (0.776). DeLong’s test was used to compare the ROC curve AUC values of random forest with the other three models (the bottom right of Figure 7). The results showed that there was no statistically significant difference in the AUC values between the random forest and nearest neighbors, RBF SVM, and QDA (*p* > 0.05). However, random forest had the highest AUC score (0.8380) and specificity (0.8077) on the test set, and its accuracy (0.7071), precision (0.7115), sensitivity (0.7071), and f1-score (0.7032) were all above 0.70, indicating excellent performance (Table 4).

In summary, the overall model built with the random forest algorithm had the best predictive performance. The two sets of data before and after the lifting of the restrictions of the pandemic were, respectively, input into the model, with AUC values of 0.76 and 0.81, showing stable performance (Figure 8).

### 3.4. Model Stability Assessment

To test the stability of the established overall model’s predictive effect, the two sets of data before and after the lifting of the restrictions of the pandemic were each randomly divided into three groups, and the overall and group data were input into the model to test the stability of the indicators of its predictive effectiveness. Since the data did not meet the normal distribution in the normality test (*p* < 0.05), non-parametric tests were used. The results indicated that there were no significant differences in the data indicators across each group (*p* > 0.05), confirming that the model remains stable regardless of whether it is applied to data before or after the lifting of pandemic restrictions, meaning it is not impacted by changes in pandemic prevention and control measures (Table 5).

### 3.5. Model Explainability Analysis

The SHAP graph importance results based on the RF model are shown in Figure 9. In the variable SHAP graph, each sample is represented by one point. The horizontal axis represents the SHAP value of the feature, which indicates the impact distribution of each feature on the model’s output. The color represents the feature value, with high values represented by red and low values represented by blue. The vertical axis reflects the feature importance ranking. The factors that have a greater impact on the delay in seeking medical treatment for oral cancer patients are as follows: the person lived with >the impact of COVID-19 > the difficulty level of seeking help from others for misunderstood hospital materials (HL2) > the difficulty level of understanding one’s health condition through written information (HL3) > checking the mouth regularly > living with others or not.

## 4. Discussion

In 1938, the concept of delay in seeking medical treatment came into being [7]. Scott SE and others summarized the results of eight similar studies and found that the occult nature of oral lesions and diagnostic delays were considered to be the cause of the high incidence of advanced oral cancer [8]. Delays in treatment before the onset of oral cancer can lead to disease progression to advanced disease, resulting in decreased survival rates [9]. Conversely, the early treatment of oral cancer has less damage and a better prognosis, with higher survival rates and quality of life for patients [10]. Fortunately, due to the discovery of a large number of risk factors, a long natural history, and the possibility of the visual inspection of lesions, the prevention of oral cancer is highly feasible [11]. This study is the first large-sample study of factors affecting the delay in seeking medical attention among oral cancer patients in western China, with high research value.

Inhabiting status has a significant impact on patient delay in seeking medical treatment. Previous research results have shown that there are significant differences in the disease burden of oral cancer by age [12], and the incidence and mortality of oral cancer gradually increase with age. Elderly people are a high-risk group for oral cancer. In this study, 42.6% of patients were over 60 years old. Population aging is a serious problem facing China, and it is expected that, by 2050, China will have 400 million elderly people over 65 years old, of which 150 million will be over 80 years old [13]. On the contrary, the number of people in the adult labor force is insufficient, which is related to the one-child policy introduced in 1980 and abolished in 2016. The way Chinese people take care of elderly people is generally dominated by children taking care of elderly people. In this study, the proportion of patients with adult children living with them who experienced delays in seeking medical attention was significantly lower than that of patients without adult children living with them, indicating a statistically significant difference. Adult children have received certain education, have good care ability and mobility, and can detect symptoms early and send patients to the hospital in time. Cohabitants are an important factor affecting patient medical behavior, which is also reflected in the SHAP graph of the risk prediction model.

With the development of urbanization, a large number of adult laborers in rural areas migrate to cities, and the population of “empty nest” elderly people in rural areas increases significantly. Elderly people lack the care and financial support of adult children [14]. These elderly people usually have more chronic diseases (two or more), and over time, they have become accustomed to and adapted to them, so they often ignore chronic symptoms and underestimate the conditions [15]. A large amount of elderly people even have to take care of the left-behind children, leaving elderly people in rural areas no time to take care of their health [16].

In response to the lack of care for “empty nest” elderly people, we suggest gradually addressing the issues of high medical costs and the insufficient coverage of medical insurance for these elderly individuals. It is essential to vigorously develop primary medical services and encourage general practitioners to serve at the grassroots level, providing them with economic and policy subsidies. Additionally, it is important to establish health records for permanent residents in the jurisdiction area and create chronic disease cards. In an oral cancer screening project conducted in Thailand, participants expressed a preference for home visits over going to the hospital themselves [17], which may be related to the older age of the patients and their relative distance from qualified hospitals. Therefore, we believe that home visit services should be fully implemented. Home visits are a supplement to outpatient services and can provide on-site examinations and treatments for elderly and physically challenged patients. The advantages of home visits also include the ability to understand and assess the patient’s social support situation on-site, allowing for targeted treatment plans [18].

The place of residence and the accessibility of medical and health services have an effect on the behavior of patients with oral cancer seeking medical treatment. Geographical location and socio-economic inequalities have a significant impact on the occurrence of oral cancer [19,20]. High-incidence areas of oral cancer are mostly located in developing countries in Asia. China is a developing country in Asia, and Guangxi is an economically underdeveloped region in the western part of China, as well as an autonomous region inhabited by many ethnic groups. It has its uniqueness in geography, economy, culture, and medical and health resources. In this study, the difference in the impact of the accessibility of medical and health services on patients’ delay in seeking medical attention was not statistically significant, which may be related to the fact that the sample was sourced from a single center. From Figure 3, we can understand the accessibility of Guangxi oral cancer patients to various levels of medical and health institutions.

In China, hospitals are divided into three levels and ten grades based on indicators such as hospital size, medical hardware equipment, talent, and technical strength. The primary institutions include community health service centers, township health centers, and village clinics. In recent years, China has made great efforts to improve the level of medical security in rural areas and reform and improve the operation mechanism of the rural medical and health system [21]. Primary medical and health service institutions have basically been popularized, and the results of this study also reflect this trend; under the mode of car travel, the accessibility level of first-level medical institutions is the highest, and urban and rural residents in Guangxi can conveniently reach primary medical and health institutions for medical treatment. However, the level of medical services provided by primary medical institutions is not optimal. In a study in Sichuan Province, the diagnostic accuracy of primary doctors in rural areas for unstable angina and type 2 diabetes was evaluated using standardized patient methods. A total of 172 rural primary doctors were included in the study, and 186 standardized patient visits were completed. The results showed that the correct diagnosis rate was only 48.39% [22]. In another survey on the cognitive level of primary doctors’ knowledge of COPD, 229 rural doctors in Zigong City, Sichuan Province, had a cognitive level of COPD knowledge of only 14.4% [23]. In addition, 73 representative rural primary medical institutions were selected in Hubei, Guangxi, and Jiangxi to investigate the use rate of antibiotics in prescriptions, and it was found that all these institutions had the problem of antibiotic abuse [24]. This study also found that doctors in primary medical institutions tend to prescribe antibiotics for initial patients with “ulcers” or swellings in the mouth rather than referring them to higher-level institutions, which may delay the diagnosis of patient’s condition [25].

Tertiary hospitals can provide high-level specialized medical and health services to several regions, and their locations are mostly in densely populated urban areas. However, oral cancer patients mostly live in remote rural areas with scarce medical resources, so their accessibility to tertiary medical institutions is poor. Oral specialty hospitals have the highest level of the diagnosis and treatment of oral cancer, but their accessibility is the worst. At present, Guangxi only has two oral specialty hospitals, located in Nanning and Guilin, the two major cities in Guangxi with relatively complete medical resources. Primary care physicians and dentists play an important role in referring oral cancer cases to specialist services. A study in Canada found that dentists are more likely than primary care physicians to detect asymptomatic cases [26]. Patients with oral cancer can first go to the oral department of a general hospital or an oral clinic for screening. The accessibility level of oral clinics is generally good, they are mostly located in densely populated urban areas and rarely in remote rural areas.

The uneven distribution of medical resources and the low level of the diagnosis and treatment of primary medical service institutions greatly affect the seeking behavior of oral cancer patients in Guangxi. To solve these problems, the government has implemented a medical consortium model, trying to promote communication and exchanges between local tertiary hospitals, secondary hospitals, and primary hospitals and accelerate the construction of primary medical information and the establishment of primary clinical pathways to improve the initial diagnosis ability of primary doctors for common diseases. Guangxi is also accelerating the construction of transportation infrastructure to improve people’s convenience in accessing corresponding medical institutions.

The impact of the COVID-19 remains after the lifting of epidemic restrictions. As this study spans before and after the lifting of the restrictions of the epidemic, there was a decline in the utilization of medical services in hospitals during this period, with most people choosing to delay seeking medical care, even in urgent situations. Reports show that the number of emergency department visits decreased during the COVID-19 pandemic [27]; 41% of Americans delayed or avoided going to the hospital for urgent or non-urgent medical care due to COVID-19 [28], and in England, the delay in diagnosis caused by COVID-19 led to a significant increase in preventable cancer mortality [29]. In this study, COVID-19 was a significant factor in the delay in seeking medical care for oral cancer patients.

Six months after China lifted the restrictions of the pandemic, in May 2023, the Director-General of the World Health Organization, Tedros Adhanom Ghebreyesus, announced that the COVID-19 pandemic is now an established and ongoing health issue and no longer constitutes a public health emergency of international concern. However, even after lifting the restrictions of the pandemic, the behavior of oral cancer patients in seeking medical care is still affected, possibly because the aftermath of the pandemic has not yet dissipated. Since the lifting of the restrictions of the pandemic, the risk of contracting COVID-19 has greatly increased, especially for high-risk groups (elderly people, those with underlying diseases, etc.), who are more likely to develop severe symptoms. There is ample evidence that, after “recovery” from COVID-19, a considerable proportion of people experience sequelae in the immune, blood, respiratory, cardiovascular, digestive, and other systems [30]. One study included 443,588 participants with a primary infection, 40,947 with two or more infections, and 5,334,729 uninfected participants [31]. The researchers found that, compared to those with only one infection, individuals with repeated infections had a 2.17 times higher risk of all-cause mortality and a 3.32 times higher risk of hospitalization. Therefore, even if oral cancer patients notice their oral health issues, they may still choose to delay seeking medical care out of fear of contracting COVID-19; another scenario is that patients who have already contracted COVID-19 may choose to go to the hospital to address more urgent symptoms (such as respiratory symptoms, cardiovascular symptoms, etc.) rather than prioritizing oral health issues. However, how long the impact of the COVID-19 pandemic will last remains to be studied in the future.

Health literacy (HL) refers to the cognitive and social skills that determine an individual’s motivation to seek, understand, and use information to promote and maintain good health [32]. This study uses the Brief Health Literacy Screen (BHLS) scale, which consists of three questions: (1) How confident are you filling out medical forms by yourself? (2) How often do you have someone (like a family member, friend, hospital/clinic worker, or caregiver) help you read hospital materials? (3) How often do you have problems learning about your medical condition or health because of difficulty understanding written information? The questionnaire uses a Likert five-point scale, where 1 represents very difficult and 5 represents no difficulty at all, with a total score of 15 points; a higher score indicates a higher level of HL. A total score of ≤9 indicates insufficient HL [33]. The advantages of this scale are its convenience and speed, and the questions are unlikely to cause patients to feel shame, sensitivity, or anxiety [34]. The scores of the three questions for patients who delay seeking medical care are lower than those of non-delayed patients. The SHAP graph of the risk prediction model shows that the difficult degree of seeking help from others for incomprehensible health information (HL2) and the difficult degree of learning about one’s medical condition through reading hospital materials (HL3) are significant factors. Among patients with insufficient health literacy, 53% are over 60 years old. There are many reasons for insufficient health literacy, including low literacy levels, visual impairments, inability to write, and lack of young family members to assist, which make it difficult for patients to obtain health information through their own efforts or support from others, making it challenging to truly understand their medical conditions, directly affecting their behavior in seeking medical care. Having good health literacy is extremely important, as it allows patients to pay attention to oral health issues and even learn self-examination, thus allowing the early detection of oral diseases, timely medical care, and timely and effective treatment.

To improve patients’ health literacy, it is essential first to popularize knowledge about oral cancer. In recent years, the government has proposed the Healthy China initiative; one of its major actions is the Health Knowledge Popularization Action, aimed at improving everyone’s health literacy, encouraging medical staff to participate in health promotion and education work, and establishing a health science popularization platform using new media. In addition, we suggest that community workers deeply communicate with families and regularly provide one-on-one health guidance for elderly people with low cultural knowledge and mobility issues, increasing their attention to diseases, early detection, and timely medical care.

This study found that, among patients who delay seeking medical attention, the proportion of those with regular oral check-up habits and those who brush their teeth more than twice a day is lower than that of non-delayed patients. People who regularly see dentists are a group that regularly undergo oral cancer screening, which undoubtedly protects high-risk groups. A study in the UK found that oral cancers detected during “routine checks” are more likely to be in the early stages (stage 1 or 2) than those detected after symptoms appear [25]. A large randomized controlled trial on oral cancer screening conducted over 15 years in India showed that oral cancer screening reduced the mortality rate of high-risk groups by 24% [35]. Patients with regular oral check-up habits pay more attention to oral problems, can detect oral diseases in time, and are more willing to seek treatment for oral cancer in hospitals. In a study by the International Head and Neck Cancer Epidemiology Consortium (INHANCE) on the role of oral hygiene in head and neck cancer, they observed a negative correlation between the occurrence of head and neck cancer (HNC) and regular tooth brushing every day [36]. Oral hygiene habits are one of the factors affecting the incidence of oral cancer and also affect patients’ behavior in seeking medical attention for oral cancer. Therefore, it is necessary to improve patients’ oral hygiene habits and behaviors for the tertiary prevention of oral cancer.

The establishment of the random forest model has predictive significance. In this study, a risk prediction model for delayed medical attention was constructed based on machine learning algorithms, and the predictive effects of four prediction models, nearest neighbors, the RBF SVM, random forest, and QDA, on the delay in seeking medical treatment for oral cancer patients in Guangxi Province were compared and analyzed. This study found that the random forest (RF) model has the best predictive performance and is stable both before and after the lifting of the pandemic restrictions. RF randomly samples the original dataset to form different sample datasets, then builds different decision tree models based on these datasets, and finally obtains the final result based on the average value (for regression models) or voting (for classification models) of these decision tree models. To ensure the generalization ability of the model, random forests usually follow two basic principles when building each tree: data randomness and feature randomness. Its advantages include high classification accuracy, the ability to identify variable importance, and the ability to perform various types of data analysis. Currently, RF is a common ensemble learning method used for classification, regression, and other tasks [37]. In addition, the SHAP diagram ranks the six factors selected according to their importance and directly explains their impact on the medical behavior of oral cancer patients, providing evidence for public health policymakers and healthcare providers to early identify high-risk groups of oral cancer and take corresponding measures.

Since December 2019, the COVID-19 pandemic has swept across the world, and pandemic prevention and control work face serious challenges at present. Due to the particularity of oral diagnosis and treatment, patients cannot wear masks during oral diagnosis and treatment, and close face-to-face contact between doctors and patients greatly increases the risk of infection. Therefore, before the lifting of the restrictions of the pandemic, oral clinics adopted a relatively strict triage system [38]. For suspected or confirmed COVID-19 or close contacts of COVID-19, treatments were postponed as much as possible until after the infection period or the end of self-isolation, which undoubtedly affected the visiting behavior of oral cancer patients. On 7 December, the state announced the “Notice on Further Optimizing and Implementing COVID-19 Prevention and Control Measures,” and after three years of COVID-19 prevention and control in China, the novel coronavirus infection returned from initial Class A management to Class B management. This study included patients who visited the hospital from June 2022 to July 2023, which was a transition period from the strict control to the gradual orderly opening of the COVID-19 pandemic, so the impact of the pandemic should be fully considered. We conducted a stability assessment of the established model and found that it has stability whether applied to data before or after the relaxation of pandemic prevention and control measures, meaning it is not affected by changes in COVID-19 pandemic prevention and control.

## 5. Limitations

This study has some limitations. The sample comes from a single-center convenience sample, making it difficult to directly verify the impact of the level of accessibility of medical and health services on the delayed medical behavior of oral cancer patients. In addition, the constructed model has not been externally validated. Future research will consider multi-center studies, and subsequent research should pay more attention to the practical application value, optimize variable settings, emphasize external validation to further optimize the model, and demonstrate the external adaptability of the model.

## 6. Conclusions

In summary, factors such as cohabitation status, being affected by the COVID-19 pandemic, the mode of transportation used to seek medical treatment, patients’ health literacy, oral examination habits, ways to obtain oral knowledge, oral hygiene habits, the distance between residence and medical institutions, etc., are factors affecting patients’ delay in seeking medical treatment. The random forest model constructed based on the data of the delayed medical treatment of oral cancer patients in this study has a good predictive performance and can be used to predict the risk of delayed medical treatment for oral cancer patients in Guangxi Province, providing a reference for public health policymakers and healthcare providers in western China to early identify high-risk groups of oral cancer and take corresponding measures.

## Figures and Tables

**Figure 1 healthcare-12-02252-f001:**
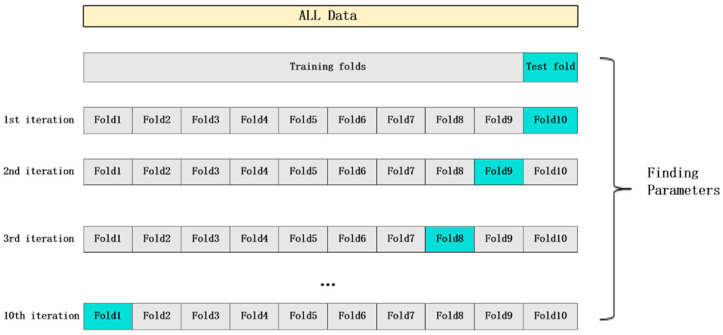
Flowchart of 10-fold cross-validation.

**Figure 2 healthcare-12-02252-f002:**
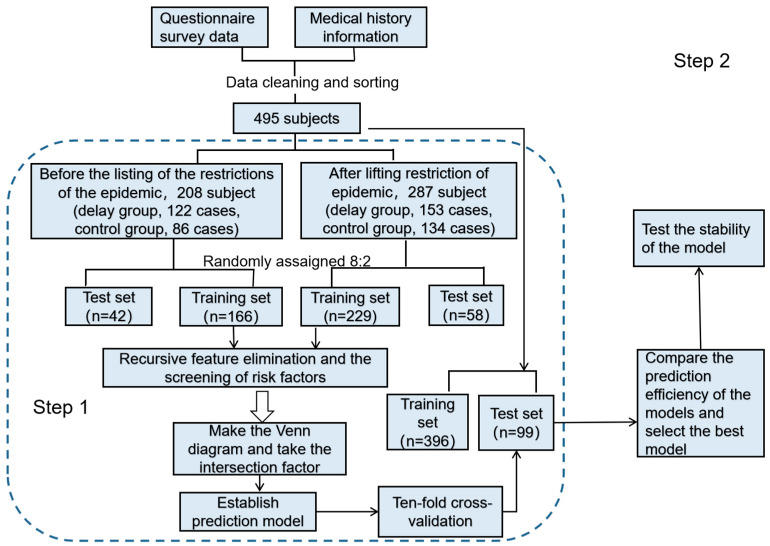
An overview of the model building process and the deep learning model architecture.

**Figure 3 healthcare-12-02252-f003:**
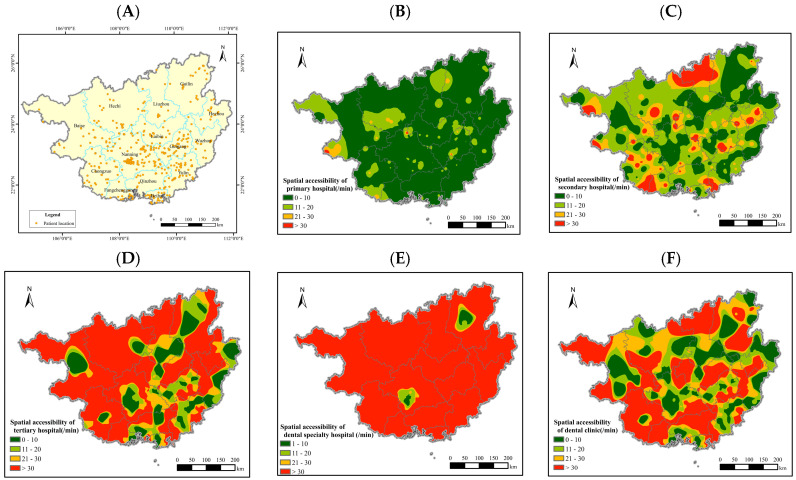
Analysis of the accessibility of medical and health services for oral cancer patients in Guangxi under the mode of automobile travel. ((**A**): Spatial distribution of residence of oral cancer patients in Guangxi; (**B**–**F**): accessibility of health services to primary, secondary, and tertiary hospitals, dental specialty hospitals, and oral clinics under the automobile travel mode).

**Figure 4 healthcare-12-02252-f004:**
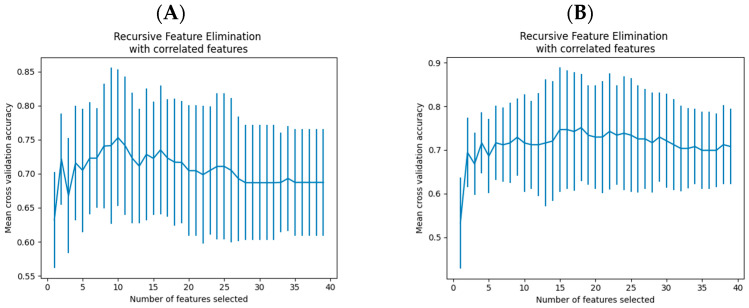
Recursive feature elimination with correlated features ((**A**): features that were screened before the lifting of pandemic restrictions; (**B**): features that were screened after the lifting of pandemic restrictions).

**Figure 5 healthcare-12-02252-f005:**
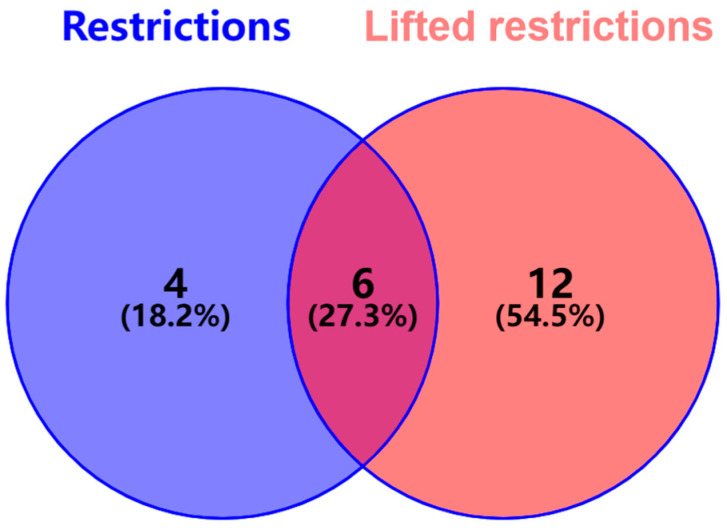
Venn diagram for analysis of common influencing factors before and after lifting of restrictions of pandemic.

**Figure 6 healthcare-12-02252-f006:**
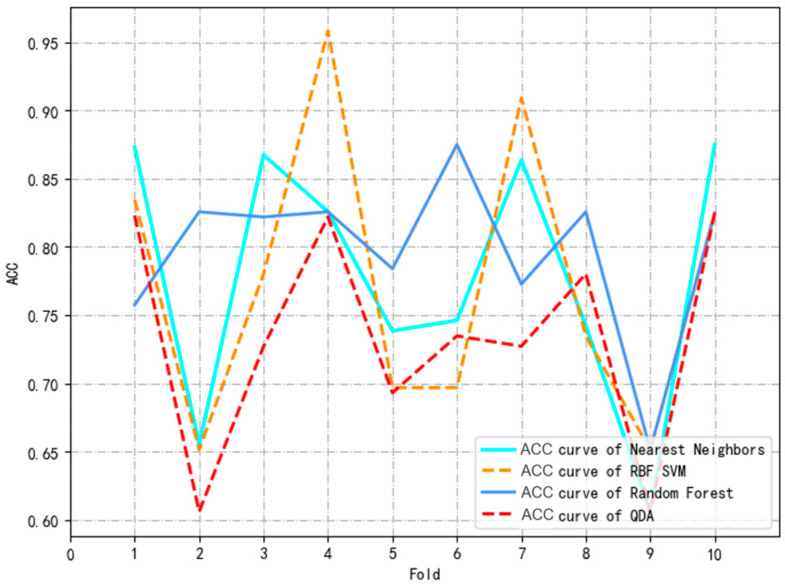
10-fold ross-validation ACC change chart.

**Figure 7 healthcare-12-02252-f007:**
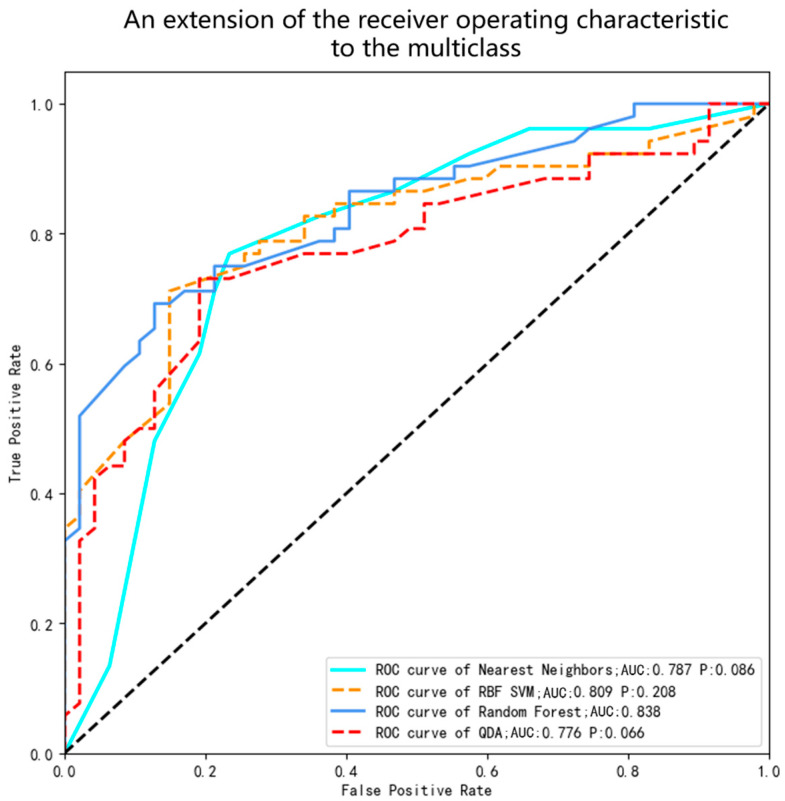
ROC curve of the prediction effectiveness of four models (nearest neighbors, the RBF SVM, random forest, and QDA) on the risk of patient delay behavior for oral cancer in Guangxi.

**Figure 8 healthcare-12-02252-f008:**
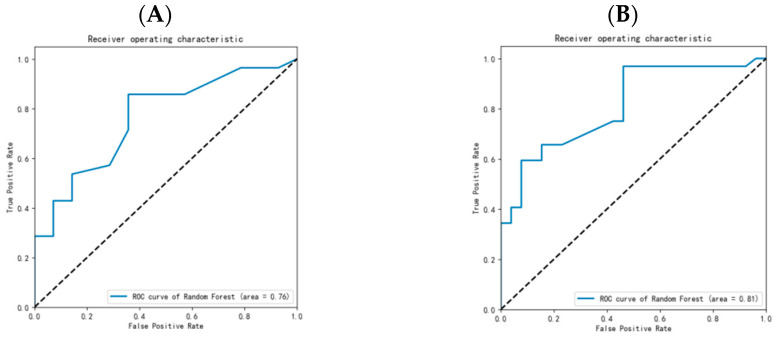
ROC curve of prediction efficiency of random forest model for delayed risk of oral cancer in Guangxi ((**A**): ROC curve before lifting of restrictions of pandemic; (**B**): ROC curve after lifting of restrictions of pandemic).

**Figure 9 healthcare-12-02252-f009:**
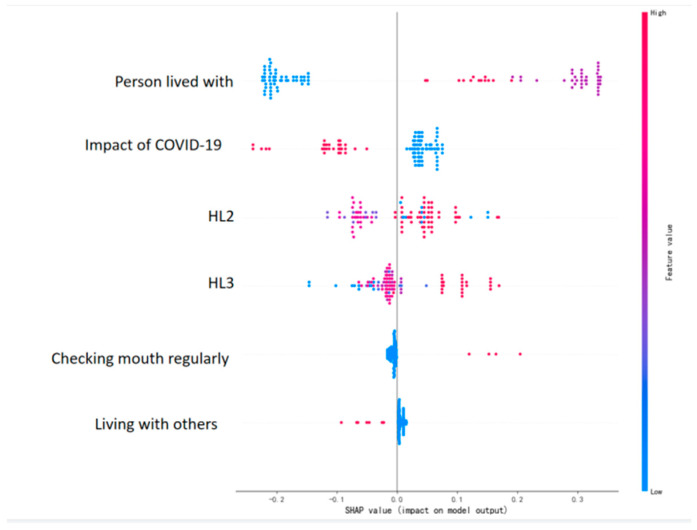
SHAP diagram of risk prediction characteristics of delayed medical treatment for oral cancer patients in Guangxi based on random forest model.

**Table 1 healthcare-12-02252-t001:** Tumor type, TNM staging, and whether there were systemic diseases.

Tumor Type	Quantity
Lip/Parotid/Floor of Mouth/Pharynx	61
Cheek	84
Tongue	239
Gum Cancer	84
Palatal Cancer	27
**TNM Stage**	
Stage 1	71
Stage 2	154
Stage 3	77
Stage 4	193
Presence of Systemic Diseases	
Yes	291
No	204
Systemic Disease Classification	
None	206
Circulatory System Diseases	133
Endocrine System Diseases	35
Others	121

**Table 2 healthcare-12-02252-t002:** Demographic characteristics of patients with oral cancer.

Demographic Data		During Strict Travel Restrictions		Lifted Restrictions		*p*
Gender	Male	130	62.50%	188	65.50%	0.491
	Female	78	37.50%	99	34.50%	
Nation	Han ethnicity	119	57.20%	173	60.30%	0.518
	Ethnic minorities	89	42.80%	114	39.70%	
Occupation	Workers	41	19.70%	85	29.60%	**0.032** *
	Farmers	88	42.30%	98	34.10%	
	Professionals and others	79	38.00%	104	36.20%	
Age	<40	17	8.20%	41	14.30%	0.113
	40–59	99	47.60%	127	44.30%	
	>60	92	44.20%	119	41.50%	
Residency	Rural	104	50.00%	149	51.90%	0.674
	Urban	104	50.00%	138	48.10%	
Educational level	Below primary education	11	5.30%	29	10.10%	0.341
	Primary school	43	20.70%	64	22.30%	
	Middle school	65	31.30%	86	30.00%	
	High school	50	24.00%	62	21.60%	
	≥College	39	18.80%	46	16.00%	
Monthly income	<1500 RMB	49	23.60%	99	34.50%	0.112
	1500–3000 RMB	57	27.40%	63	22.00%	
	3001–5000 RMB	62	29.80%	81	28.20%	
	5001–8000 RMB	31	14.90%	34	11.80%	
	>8000 RMB	9	4.30%	10	3.50%	
Marital status	Single	9	4.30%	30	10.50%	**0.007** *
	Married	184	88.50%	220	76.70%	

* Statistically signifcant.

**Table 3 healthcare-12-02252-t003:** Multivariate analysis of delayed medical treatment in patients with oral cancer.

	Restrictions		Lifted Restrictions	
Parameter	Multivariate Analysis		Multivariate Analysis	
	OR (95%CI)	*p*	OR (95%CI)	*p*
Person lived with (Adult children vs. Spouse)	41.99 (10.71~164.56)	**<0.001** *	24.72 (10.11~60.45)	**<0.001** *
Person lived with (Adult children vs. No one or others)	9.09 (1.29~63.84)	**0.026** *	5.77 (2.15~15.53)	**0.001** *
Vehicle (On foot vs. Electric bicycle)	5.65 (1.33~23.99)	**0.028** *	3.09 (1.08~8.84)	**0.035** *
Impact of COVID-19 (Yes vs. No)	8.66 (2.28~32.82)	**0.002** *	-	**-**
Access to knowledge (Network and others vs. Relatives and friends)	14.83 (1.83~120.55)	**0.012** *	-	**-**
Checking mouth regularly (No vs. Yes)	26.35 (2.86~242.80)	**0.004** *	-	**-**
Distance to nearest medical facility (<2 km vs. 2–5 km)	-	**-**	2.94 (1.04~8.34)	**0.043** *
Distance to nearest medical facility (<2 km vs. 6–10 km)	-	**-**	4.32 (1.03~18.09)	**0.045** *
Chewing hard objects	-	**-**	2.15 (1.05~4.38)	**0.035** *

* Statistically signifcant.

**Table 4 healthcare-12-02252-t004:** Comparison of comprehensive performance of machine learning prediction models.

	Random Forest	Nearest Neighbors	RBF SVM	QDA
test set				
accuracy	0.7071	0.7677	0.7273	0.6566
AUC score	**0.8380** *	0.7866	0.8089	0.7762
sensitivity	0.7071	0.7677	0.7273	0.6566
specificity	**0.8077** *	0.7692	0.7885	0.8077
precision	0.7115	0.768	0.7282	0.6651
f1-score	0.7032	0.7678	0.7259	0.6469

* Statistically signifcant.

**Table 5 healthcare-12-02252-t005:** Comparison of comprehensive performance stability of prediction models.

	Group A	Group A (1)	Group A (2)	Group A (3)	Data B	Group B (1)	Group B (2)	Group B (3)	Z	*p*
*n*	208	69	69	70	287	96	96	95		
test set										
accuracy	0.7619	0.7857	0.7857	0.7143	0.7071	0.6500	0.6500	0.7368	−2.045	0.057
AUC score	0.7628	0.7708	0.7917	0.9271	0.8125	0.6700	0.6700	0.7670	−1.1620	0.3140
sensitivity	0.7619	0.7857	0.7857	0.7143	0.7414	0.6500	0.6500	0.7368	−1.7530	0.1140
specificity	0.8214	0.8750	0.8333	0.6250	0.6562	0.7000	0.6667	0.7500	−1.155	0.343
precision	0.7619	0.7873	0.7959	0.7440	0.7623	0.6515	0.7188	0.7439	−1.7320	0.1140

(Group A: data before lifting of restrictions of pandemic; Group B: data after lifting of restrictions of pandemic).

## Data Availability

The data that support the findings of this study are available from the Hospital of Stomatology, Guangxi Medical University, but restrictions apply to the availability of these data, which were used under license for the current research, and so are not publicly available. Data are however available from the authors upon reasonable request and with the permission of the Hospital of Stomatology, Guangxi Medical University. The datasets generated during and/or analyzed during the current study are available from the corresponding author on reasonable request.

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
