# Peer review of "Exploring Factors Influencing Patient Delay Behavior in Oral Cancer: The Development of a Risk Prediction Model in Western China"

_healthcare, 2024, doi:10.3390/healthcare12222252_

Round 1

Reviewer 1 Report

Comments and Suggestions for Authors

Dear editor and authors,

This is a non well-prepared study which explore factors influencing patient  with delay behavior in Oral Cancer.  This study is not innovative enough and lack of clinical significance for oncologists' treatment and decision making. It has limited relevance for preventive medicine practitioners, informing readers of the factors that may lead to treatment delays by analysing the factors that delay oral oncology treatment, but does not provide a solution to the problem of treatment delays. Here some major revisions considerations are necessary, as follows:

Comments 1:

Line 103

In this study, the delay in seeking medical treatment for oral cancer was defined as 103

the time from the onset of symptoms to the first visit > 3 months.

Is there a relevant clinical basis for considering 3 months as the judgement standard for deferred treatment?

Comments 2:

Line 105

What is the rationale for the authors' choice of these variables? Are there other factors that influence patients to defer treatment?

Comments 3:

Different tumours may present differently, and the authors do not provide what type of tumour these 495 patients had, and the stage of the tumour, whether it was a primary tumour or a recurrent tumour? Were there other comorbidities, and associated systemic diseases?

Comments 4:

The authors lacked follow-up information on this study population; were these patients with deferred tumours treated? What kind of treatment was administered? What was the prognosis after treatment?

Comments 5:

line 530: The study was conducted before and after lifting the restrictions of the epidemic, consisting of two sets of data, but the epidemic has now passed, which may affect extrapolation.

Just as the author mentioned in the discussion, this dataset included confounding factor which influence the outcome of this research.

Comments 6:

Line 173

The authors apply deep learning to build predictive models, yet there is no analysis or discussion in the conclusion and discussion, leaving the reader feeling puzzled. The predictive model for deep learning is puzzling and abrupt in the article.

Comments on the Quality of English Language

Dear editor and authors,

This is a non well-prepared study which explore factors influencing patient  with delay behavior in Oral Cancer.  This study is not innovative enough and lack of clinical significance for oncologists' treatment and decision making. It has limited relevance for preventive medicine practitioners, informing readers of the factors that may lead to treatment delays by analysing the factors that delay oral oncology treatment, but does not provide a solution to the problem of treatment delays. Here some major revisions considerations are necessary, as follows:

Comments 1:

Line 103

In this study, the delay in seeking medical treatment for oral cancer was defined as 103

the time from the onset of symptoms to the first visit > 3 months.

Is there a relevant clinical basis for considering 3 months as the judgement standard for deferred treatment?

Comments 2:

Line 105

What is the rationale for the authors' choice of these variables? Are there other factors that influence patients to defer treatment?

Comments 3:

Different tumours may present differently, and the authors do not provide what type of tumour these 495 patients had, and the stage of the tumour, whether it was a primary tumour or a recurrent tumour? Were there other comorbidities, and associated systemic diseases?

Comments 4:

The authors lacked follow-up information on this study population; were these patients with deferred tumours treated? What kind of treatment was administered? What was the prognosis after treatment?

Comments 5:

line 530: The study was conducted before and after lifting the restrictions of the epidemic, consisting of two sets of data, but the epidemic has now passed, which may affect extrapolation.

Just as the author mentioned in the discussion, this dataset included confounding factor which influence the outcome of this research.

Comments 6:

Line 173

The authors apply deep learning to build predictive models, yet there is no analysis or discussion in the conclusion and discussion, leaving the reader feeling puzzled. The predictive model for deep learning is puzzling and abrupt in the article.

Author Response

We would like to thank the reviewers for their insightful and constructive comments.

We have now fully addressed the comments and revised the manuscript accordingly.

For convenience, the response to the reviewers’ comments are shown in blue in the

response letter. In the revised manuscript, the key points that we made to address the

reviewers’ comments are shown in blue.

Comments 1:

Line 103

In this study, the delay in seeking medical treatment for oral cancer was defined as 103

the time from the onset of symptoms to the first visit > 3 months.

Is there a relevant clinical basis for considering 3 months as the judgement standard for deferred treatment?

Response: Thank you very much for your positive comments and suggestions. Patient delay refers to the time interval between when a patient first notices symptoms and when they first consult a healthcare professional about those symptoms. Allison et al1. found that delays longer than three months pose a risk of the disease progressing to an advanced stage, but daily screening and early diagnosis of oral cancer are not easy. Although most oral cancer patients do not have obvious symptoms in the early stages, dentists can detect them during routine oral examinations. Early detection of these diseases helps reduce the morbidity and mortality associated with cancer. Therefore, raising awareness about regular oral check-ups will help with the early diagnosis of oral cancer and minimize its incidence. Patient delay is the most important factor in the delay pathway. Currently, there is limited research on patient delay in China, and no reports have been published in the western region. This study aims to fill this gap.

Reference:

  1. Allison, P., E. Franco, and J. Feine, Predictors of professional diagnostic delays for upper aerodigestive tract carcinoma. Oral Oncol, 1998. 34(2): p. 127-32

Comments 2:

Line 105

What is the rationale for the authors' choice of these variables? Are there other factors that influence patients to defer treatment?

Response: Thank you very much for your positive comments and suggestions. There is extensive research on the factors influencing delayed medical treatment behavior in oral cancer patients abroad. Scholars have found that the relationship between oral cancer patients' delayed medical treatment behavior and their socio-demographic variables, such as gender, age, and marital status, is not significant. However, it is closely related to the patient's educational level1, cognition of oral cancer2, psychological and social variables3, socioeconomic level, and oral examination habits. In addition, social support factors and the distance between the place of residence and medical institutions also have a significant impact on the delayed medical treatment behavior of oral cancer. Based on previous research, this study has newly discovered that personal health literacy and the situation of cohabitants are also related.

Reference:

  1. NOONAN B. Understanding the reasons why patients delay seeking treatment for oral cancer symptoms from a primary health care professional: an integrative literature review [J]. European journal of oncology nursing : the official journal of European Oncology Nursing Society, 2014, 18(1): 118-24.
  2. KERDPON D, JANTHARAPATTANA K, SRIPLUNG H. Factors related to diagnostic delay of oral squamous cell carcinoma in southern Thailand: Revisited [J]. Oral diseases, 2018, 24(3): 347-54.
  3. TROMP D M, BROUHA X D, DE LEEUW J R, et al. Psychological factors and patient delay in patients with head and neck cancer [J]. European journal of cancer (Oxford, England : 1990), 2004, 40(10): 1509-16.

Comments 3:

Different tumours may present differently, and the authors do not provide what type of tumour these 495 patients had, and the stage of the tumour, whether it was a primary tumour or a recurrent tumour? Were there other comorbidities, and associated systemic diseases?

Response: Thanks for the comments. We have provided a table that includes tumor type, TNM staging, and whether there are systemic diseases.

Comments 4:

The authors lacked follow-up information on this study population; were these patients with deferred tumours treated? What kind of treatment was administered? What was the prognosis after treatment?

Response: Thanks for the comments. The research purpose of this project is to study the behavior of delayed diagnosis and treatment, rather than to study the prognosis. Therefore, no data on how the patients' prognosis was after treatment was collected.

Comments 5:

line 530: The study was conducted before and after lifting the restrictions of the epidemic, consisting of two sets of data, but the epidemic has now passed, which may affect extrapolation.

Just as the author mentioned in the discussion, this dataset included confounding factor which influence the outcome of this research.

Response: Thanks for the comments. This model has taken into account the confounding factors caused by the epidemic and conducted analysis and discussion. It was found that the model also has good predictive effects under epidemic conditions and is not affected by the broader environment of the epidemic.

Comments 6:

Line 173

The authors apply deep learning to build predictive models, yet there is no analysis or discussion in the conclusion and discussion, leaving the reader feeling puzzled. The predictive model for deep learning is puzzling and abrupt in the article.

Response: Thanks for the comments. We had add analysis in the introduction in line .

Reviewer 2 Report

Comments and Suggestions for Authors

I congratulate the authors on their work. I believe the manuscript is acceptable if the following correction suggestions are made.

- Provide more information about the implications of patient delayed behavior in the context of oral cancer in the introduction.

- Clearly state the purpose of your study and your null hypothesis in the last paragraph of the introduction. Clearly state the rejection/acceptance of your null hypothesis in the first paragraph of the discussion.

- What do you mean by oral cancer? Which types of cancer were classified as oral cancer and why? What was the basis for this classification? Visual/clinical examination or pathological/histological examination? Provide detailed information about the identification and typing of oral cancers in a separate paragraph in the Methods section.

- What is the relevance of the ethics approval statement to statistical analysis? Remove it from this subheading.

- Rewrite p's of 0.000 in relevant text and tables as <0.001.

- The entire text should be checked and edited for spelling, grammar, and punctuation. Numerous errors are noted.

Comments on the Quality of English Language

- The entire text should be checked and edited for spelling, grammar, and punctuation. Numerous errors are noted.

Author Response

We would like to thank the reviewers for their insightful and constructive comments.

We have now fully addressed the comments and revised the manuscript accordingly.

For convenience, the response to the reviewers’ comments are shown in blue in the

response letter. In the revised manuscript, the key points that we made to address the

reviewers’ comments are shown in blue.

Provide more information about the implications of patient delayed behavior in the context of oral cancer in the introduction.

Response: Thanks for the comments. We had add more information about the implications of patient delayed behavior in the context of oral cancer in the introduction

- Clearly state the purpose of your study and your null hypothesis in the last paragraph of the introduction. Clearly state the rejection/acceptance of your null hypothesis in the first paragraph of the discussion.

Response: Thanks for the comments. We had state the purpose of our study in the last paragraph of the introduction.

- What do you mean by oral cancer? Which types of cancer were classified as oral cancer and why? What was the basis for this classification? Visual/clinical examination or pathological/histological examination? Provide detailed information about the identification and typing of oral cancers in a separate paragraph in the Methods section.

Response: Thanks for the comments. We had add more detailed information about the identification and typing of oral cancers in the Methods section.

- What is the relevance of the ethics approval statement to statistical analysis? Remove it from this subheading.

Response: Thanks for the comments. We had Remove “the relevance of the ethics approval statement to statistical analysis” from this subheading.

- Rewrite p's of 0.000 in relevant text and tables as <0.001.

Response: Thanks for the comments. We had rewrite p's of 0.000 as <0.001.

- The entire text should be checked and edited for spelling, grammar, and punctuation. Numerous errors are noted.

Response: Thanks for the comments. We had checked and edited for spelling, grammar, and punctuation in the entire text.

Comments on the Quality of English Language

- The entire text should be checked and edited for spelling, grammar, and punctuation. Numerous errors are noted.

Response: Thanks for the comments. We had checked and edited for spelling, grammar, and punctuation in the entire text.

Reviewer 3 Report

Comments and Suggestions for Authors

Introduction is not clear. Authors have not mentioned why is the study important and most importantly Aim and objective of study is not mentioned in introduction section. references are not marked properly. 

Material and method: This part should be rearraged and prediction model should come after research method. this will make it easier to understand. 

Results: it is well presented. 

Discussion: try to compare more studies with the result of this study. it is important to describe how the prediction model is helpful and have these model being utilized in other studies as well. 

Conclusion: write according to the result of the study 

Comments on the Quality of English Language

English editing is required. 

Author Response

We would like to thank the reviewers for their insightful and constructive comments.

We have now fully addressed the comments and revised the manuscript accordingly.

For convenience, the response to the reviewers’ comments are shown in blue in the

response letter. In the revised manuscript, the key points that we made to address the

reviewers’ comments are shown in blue.

Introduction is not clear. Authors have not mentioned why is the study important and most importantly Aim and objective of study is not mentioned in introduction section. references are not marked properly.

Response: Thanks for the comments. In this study, to explore new factors related to the delayed medical treatment behavior of oral cancer patients, we investigated various aspects including the patients' demographic information, social support status, site of onset, the impact of the COVID-19, personal health literacy, and the accessibility level of medical and health services in the patients' place of residence. We also established a new predictive model that can avoid the interference of the epidemic and accurately predict the risk factors for delayed medical treatment in oral cancer patients. This model can be applied to guide the formulation of public health policies, improve the medical treatment behavior of oral cancer patients, and enhance patient survival rates.

Material and method: This part should be rearraged and prediction model should come after research method. this will make it easier to understand.

Response: Thanks for the comments. We had revise the Materials and Methods section.

Results: it is well presented.

Discussion: try to compare more studies with the result of this study. it is important to describe how the prediction model is helpful and have these model being utilized in other studies as well.

Thanks for the comments. We had add more studies compare with the result of this study. Such as study found that among patients who delay seeking medical attention, the proportion of those with regular oral check-up habits and those who brush their teeth more than twice a day is lower than that of non-delayed patients. People who regularly see dentists are a group that regularly undergoes oral cancer screening, which undoubtedly protects high-risk groups. A study in the UK found that oral cancers detected during "routine checks" are more likely to be in the early stages (stage 1 or 2) than those detected after symptoms appear1. A large randomized controlled trial on oral cancer screening conducted over 15 years in India showed that oral cancer screening reduced the mortality rate of high-risk groups by 24%2. Patients with regular oral check-up habits pay more attention to oral problems, can detect oral diseases in time, and are more willing to seek treatment for oral cancer in hospitals. In a study by the International Head and Neck Cancer Epidemiology (INHANCE) Consortium on the role of oral hygiene in head and neck cancer, they observed a negative correlation between the occurrence of head and neck cancer (HNC) and regular tooth brushing every day3. Oral hygiene habits are one of the factors affecting the incidence of oral cancer and also affect the patient's behavior in seeking medical attention for oral cancer. Therefore, it is necessary to improve the patient's oral hygiene habits and behaviors for the tertiary prevention of oral cancer.

Reference:

  1. Langton, S., et al., Comparison of primary care doctors and dentists in the referral of oral cancer: a systematic review. Br J Oral Maxillofac Surg, 2020. 58(8): p. 898-917.
  2. Sankaranarayanan, R., et al., Effect of screening on oral cancer mortality in Kerala, India: a cluster-randomised controlled trial. Lancet, 2005. 365(9475): p. 1927-33.
  3. Hashim, D., et al., The role of oral hygiene in head and neck cancer: results from International Head and Neck Cancer Epidemiology (INHANCE) consortium. Ann Oncol, 2016. 27(8): p. 1619-25.

Conclusion: write according to the result of the study

Thanks for the comments.We had rewritten the conclusions based on the results.

Comments on the Quality of English Language

English editing is required.

Reviewer 4 Report

Comments and Suggestions for Authors

Exploring Factors Influencing Patient Delay Behavior in Oral Cancer: Development of a Risk Prediction Model in Western China

Journal: Healthcare MDPI

Reviewer’s comments:

Thank you for the opportunity to review this interesting manuscript. The study is well-constructed and meaningful, and the manuscript is clearly written. I have a few comments about clarity; all will be numerated below.

1.      Abstract, page 1, line 19: This phrase “delayed seeking medical treatment before lifting restrictions of the epidemic,” sounds like the patients will be the ones lifting the pandemic restrictions. Perhaps it would be better to say, “before pandemic restrictions were lifted”, or the like.

2.      General: Frequently throughout the manuscript, you refer to the pandemic as an epidemic, which is not accurate.

Introduction

3.      Page 2, line 67: In reference to Beijing and Jilin, it would be helpful for readers in other countries to know more about Jilin. Most will know that Beijing is a large city, but Jilin isn’t familiar. Is it also metropolitan? Just a short descriptive phrase would be helpful for foreign readers.

4.      Page 2, line 76: Concerning the phrase, “the epidemic prevention and control work has faced serious challenges”. While the work of prevention and control did face serious challenges, the remainder of the paragraph leads me to think that “presents” would more accurately portray your meaning. The prevention and control efforts presented challenges to both those trying to see patients with oral diagnosis, and for patients in need of oral health treatment.

5.      Page 2, lines 84-85: With regard to the following phrase: “after three years of COVID-19 prevention and control in China, the novel coronavirus infection returned from initial Class A management to Class B management”, a brief statement about what Class A and Class B management mean would help foreign readers correctly interpret the writing. I assume that Class A management means more severe restrictions, and Class B means less restrictive management. Just a descriptive phrase would help clarify.

Materials and Methods

This section is clear and well-written.

Results

6.      General: This section is necessarily dense, but well-organized.

7.      General, Figures: Most figures are not of a high enough resolution to be clearly read on a computer screen, even when enlarged, or on paper. Please use sharper image quality so that the data can be clearly interpreted.

8.      Table 2, page 7, last line: There is a formatting error in the last line of this table.

9.      Page 8, line 257: Please define health literacy before using the abbreviation. You introduce the phrase in the Methods section, but do not provide the abbreviation, nor the breakdown of the HL categories, until this section of the Results where you report the abbreviation, then much later (page 14) in the discussion, you provide an excellent description of the HL component of the data.

Discussion

10.  Page 12, line 338: The values on this line would read more clearly if stated in reverse order, as follows: “In this study, 42,6% of patients were over 60 years old.”

11.  Page 13, line 395: Regarding the sentence: “However, the level of medical services provided 394 by primary medical institutions is not optimistic,” it seems like “optimal” would be a more appropriate word than “optimistic” in this case.

12.  Page 13, general: The paragraph that takes up most of page 13 is very long. Consider breaking it into two paragraphs by starting a new paragraph at line 407, with the sentence beginning “Tertiary hospitals”.

13.  Page 13, line 417-419: These last two sentences do not hang together well. The last one is a run-on sentence and needs to be divided. It also appears to contradict itself. If oral clinics are mostly located in densely populated urban areas, how are they more accessible to the rural remote areas you are focused on in this study?

14.  Page 14, first paragraph: “Impact” does not need to be capitalized in this sentence. Also watch for using “epidemic” instead of “pandemic” in this section.

15.  Page 14, line 459: The subject/verb agreement in the first sentence of this paragraph isn’t quite right. Consider revising. Perhaps something along the lines of: “Health literacy (HL) refers to the cognitive and social skills…”.

16.  Page 14, last paragraph: You provide a detailed explanation of the health literacy data here. At least part of this paragraph might be better placed in the Methods section, so that the reader can better understand the HL-related results components. Then, in this section, just discuss the HL-related results in terms of other literature or the like. The first paragraph on page 15 could be combined with a shorter commentary on HL results with good effect.

17.  Limitations, general: Thank you for providing good notes about what further study could be done to expand on your efforts here.

Comments on the Quality of English Language

Only minor editing needed

Author Response

We would like to thank the reviewers for their insightful and constructive comments.

We have now fully addressed the comments and revised the manuscript accordingly.

For convenience, the response to the reviewers’ comments are shown in blue in the

response letter. In the revised manuscript, the key points that we made to address the

reviewers’ comments are shown in blue.

  1.      Abstract, page 1, line 19: This phrase “delayed seeking medical treatment before lifting restrictions of the epidemic,” sounds like the patients will be the ones lifting the pandemic restrictions. Perhaps it would be better to say, “before pandemic restrictions were lifted”, or the like.

Response: Thanks for the comments. We had rewrite that phrase.

  1. General: Frequently throughout the manuscript, you refer to the pandemic as an epidemic, which is not accurate.

Response: Thanks for the comments. We had change epidemic into pandemic in all the text.

Introduction

  1. Page 2, line 67: In reference to Beijing and Jilin, it would be helpful for readers in other countries to know more about Jilin. Most will know that Beijing is a large city, but Jilin isn’t familiar. Is it also metropolitan? Just a short descriptive phrase would be helpful for foreign readers.

Response: Thanks for the comments. We had add a short descriptive phrase about jilin.

  1.      Page 2, line 76: Concerning the phrase, “the epidemic prevention and control work has faced serious challenges”. While the work of prevention and control did face serious challenges, the remainder of the paragraph leads me to think that “presents” would more accurately portray your meaning. The prevention and control efforts presented challenges to both those trying to see patients with oral diagnosis, and for patients in need of oral health treatment.

Response: Thanks for the comments. We had rewrite this phrase.

  1.      Page 2, lines 84-85: With regard to the following phrase: “after three years of COVID-19 prevention and control in China, the novel coronavirus infection returned from initial Class A management to Class B management”, a brief statement about what Class A and Class B management mean would help foreign readers correctly interpret the writing. I assume that Class A management means more severe restrictions, and Class B means less restrictive management. Just a descriptive phrase would help clarify.

Response: Thanks for the comments. We had rewrite this phrase.

Materials and Methods

This section is clear and well-written.

Results

  1.      General: This section is necessarily dense, but well-organized.

Response: Thanks for the comments. We had rewrite the results.

  1.      General, Figures: Most figures are not of a high enough resolution to be clearly read on a computer screen, even when enlarged, or on paper. Please use sharper image quality so that the data can be clearly interpreted.

Response: Thanks for the comments. We had change all the figures and make it more clearly.

  1. Table 2, page 7, last line: There is a formatting error in the last line of this table.

Response: Thanks for the comments. We had change formatting error in the last line of this table3.

  1.      Page 8, line 257: Please define health literacy before using the abbreviation. You introduce the phrase in the Methods section, but do not provide the abbreviation, nor the breakdown of the HL categories, until this section of the Results where you report the abbreviation, then much later (page 14) in the discussion, you provide an excellent description of the HL component of the data.

Response: Thanks for the comments. We had added health literacy’s abbreviation in the Methods section in line 134.

Discussion

  1.  Page 12, line 338: The values on this line would read more clearly if stated in reverse order, as follows: “In this study, 42,6% of patients were over 60 years old.”

Response: Thanks for the comments. We had change into “In this study, 42,6% of patients were over 60 years old.”

  1. Page 13, line 395: Regarding the sentence: “However, the level of medical services provided 394 by primary medical institutions is not optimistic,” it seems like “optimal” would be a more appropriate word than “optimistic” in this case.

Response: Thanks for the comments. We had change “optimistic” into “optimal”.

  1.  Page 13, general: The paragraph that takes up most of page 13 is very long. Consider breaking it into two paragraphs by starting a new paragraph at line 407, with the sentence beginning “Tertiary hospitals”.

Response: Thanks for the comments. We had broken it into two paragraphs by starting a new paragraph.

  1.  Page 13, line 417-419: These last two sentences do not hang together well. The last one is a run-on sentence and needs to be divided. It also appears to contradict itself. If oral clinics are mostly located in densely populated urban areas, how are they more accessible to the rural remote areas you are focused on in this study?

Response: Thanks for the comments. According to the national conditions of China's medical treatment, it is difficult for people in remote rural areas to reach dental clinics, but they can go to primary hospitals (mostly distributed in remote rural areas).

  1. Page 14, first paragraph: “Impact” does not need to be capitalized in this sentence. Also watch for using “epidemic” instead of “pandemic” in this section.

Response: Thanks for the comments. We had revise this mistake.

  1. Page 14, line 459: The subject/ verb agreement in the first sentence of this paragraph isn’t quite right. Consider revising. Perhaps something along the lines of: “Health literacy (HL) refers to the cognitive and social skills…”.

Response: Thanks for the comments. We had revise to “Health literacy (HL) refers to the cognitive and social skills…”.

  1.  Page 14, last paragraph: You provide a detailed explanation of the health literacy data here. At least part of this paragraph might be better placed in the Methods section, so that the reader can better understand the HL-related results components. Then, in this section, just discuss the HL-related results in terms of other literature or the like. The first paragraph on page 15 could be combined with a shorter commentary on HL results with good effect.

Response: Thanks for the comments, modification has been made.

  1.  Limitations, general: Thank you for providing good notes about what further study could be done to expand on your efforts here.

Response: Thanks for the comments. We had revise limitations.

Round 2

Reviewer 1 Report

Comments and Suggestions for Authors

not qualified for publication

Comments on the Quality of English Language

not qualified for publication

Author Response

 Thanks for the comments. Although this study is not perfect, there are still many virtue in the significance and innovation of the study. This study is the first large-sample research on the risk factors of delayed medical-seeking behavior among oral cancer patients in western China. In addition, this study used machine learning algorithm to find many previously undiscovered factors affecting the delayed behavior of oral cancer patients. The discovery of these factors undoubtedly provides an important reference for the intervention of delayed behavior in oral cancer patients.

Reviewer 3 Report

Comments and Suggestions for Authors

Dear Authors,

Thank you for revising the manuscript. however there are few points that need to be considered. 

Abstract: Conclusion needs to be more precise.

Introduction: Need of the study is still not clear. In the intial version you lacked the explaination of machine learning and in this version there is no clearity on why is this study important and what exactly is prediction model. 

Author Response

comments 1:

Thank you for revising the manuscript. however there are few points that need to be considered.

Abstract: Conclusion needs to be more precise.

Response1: Thanks for the comments. We had rewrite conclusion. This study systematically elucidates the critical factors influencing patient delay behavior in oral cancer diagnosis and treatment, employing a comprehensive risk prediction model that identifies individuals at elevated risk of delay. It represents the pioneering large-scale investigation conducted in western China, focusing explicitly on the multifaceted factors affecting the delayed medical treatment behavior of oral cancer patients. The findings underscore the imperative of implementing early intervention strategies tailored to mitigate these delays. Furthermore, the study emphasizes the pivotal role of robust social support systems and positive family dynamics in facilitating timely access to healthcare services for oral cancer patients, thereby potentially improving outcomes and survival rates.

comments 2: Introduction: Need of the study is still not clear. In the intial version you lacked the explaination of machine learning and in this version there is no clearity on why is this study important and what exactly is prediction model.

Response: Thanks for the comments. We have described the implications of machine learning prediction models in more detail: the machine learning predictive model is a tool that utilizes computer algorithms to learn model parameters from training data, which are then used to predict and analyze unknown data. These models are capable of predicting a continuous outcome or classification label based on the input data features. The construction of predictive models typically involves selecting appropriate machine learning algorithms adjusting the model parameters through a training dataset to achieve optimal predictive performance. The performance of the model is usually evaluated using metrics such as accuracy, precision, and recall, to ensure that the model has good generalization ability and stability.

Furthermore, we elaborate the necessity of the study in more detail in the last paragraph of the introductionIntroduction: the significance of this study lies in: this study is the first large-sample research on the risk factors of delayed medical-seeking behavior among oral cancer patients in western China. In order to explore new factors related to the delayed medical treatment behavior of oral cancer patients, we investigated various aspects including the patients' demographic information, social support status, site of onset, the impact of the COVID-19, personal health literacy, and the accessibility level of medical and health services in the patients' place of residence. We also established a new predictive model that can avoid the interference of the pandemic and accurately predict the risk factors for delayed medical treatment in oral cancer patients. This model can be applied to guide the formulation of public health policies, improve the medical treatment behavior of oral cancer patients, and enhance patient survival rates.